# Adenoma-Derived Organoids for Precision Therapy

Tamar Evron-Levy [1], Michal Caspi [1], Amnon Wittenstein [1], Yamit Shorer-Arbel [1], Olga Shomron [2], Koret Hirschberg [2], Revital Kariv [3,†] and Rina Rosin-Arbesfeld [1,*,†]

1   Department of Clinical Microbiology and Immunology, Sackler Faculty of Medicine, Tel Aviv University, Tel Aviv 6997801, Israel; evtamron@gmail.com (T.E.-L.); mcaspi@tauex.tau.ac.il (M.C.); wittenstein@mail.tau.ac.il (A.W.); yamitsho1993@gmail.com (Y.S.-A.)
2   Department of Pathology, Sackler Faculty of Medicine, Tel Aviv University, Tel Aviv 6997801, Israel; olgashom@post.tau.ac.il (O.S.); koty@tauex.tau.ac.il (K.H.)
3   Department of Gastroenterology, Tel Aviv Sourasky Medical Center, Tel Aviv 6423906, Israel; revitalk@tlvmc.gov.il
*   Correspondence: arina@tauex.tau.ac.il
†   These authors contributed equally to this work.

**Abstract:** Human colonic organoids derived from adult tissue biopsies are based on the ability of isolated somatic epithelial stem cells to reconstitute the structure and function of the colon, offering new opportunities for studying the biology of the large intestine in both health and disease. These colonoids may also function as efficient platforms for drug screening and discovery. Here, we describe the establishment of human colonic organoids derived from healthy, and adenomatous polyp tissues. We then demonstrate that organoids grown from adenomas of familial adenomatous polyposis (FAP) patients harboring nonsense mutations in the tumor suppressor gene *adenomatous polyposis coli* (APC), can be used to establish a personalized therapeutic strategy which relies on nonsense mutation readthrough therapy.

**Keywords:** adenoma; adenomatous polyposis coli (APC); familial adenomatous polyposis (FAP)

## 1. Introduction

Colorectal cancer (CRC) is the third leading malignancy and the second most common cause of cancer-related mortalities worldwide [1]. The majority of CRC cases occur sporadically, although inherited cancer-predisposing mutations are responsible for 5–10% of all cases, and up to 30% may have familial components that can be associated with shared lifestyle and environmental factors [2,3]. Specific genetic mutations have been implicated in hereditary CRC, and predominantly in the polyposis syndromes that are associated with multiple CRC precursor colonic polyps [4]. Conditions associated with colonic adenomatous polyposis include familial adenomatous polyposis (FAP) and MUTYH-associated polyposis (MAP). Lynch syndrome, on the other hand, is generally considered a hereditary nonpolyposis colorectal cancer (HNPCC), characterized by microsatellite instability (MSI) as a consequence of a defective DNA mismatch repair (MMR) system [5]. FAP is an autosomal dominant disorder resulting from pathogenic germline mutations in the *adenomatous polyposis coli* (APC) gene. FAP patients tend to develop multiple adenomas at young age, which then progress to CRC if not removed [6]. Somatic APC mutations are also found in the vast majority of sporadic CRC cases and are considered to be one of the main triggers for cancer development [7]. Carcinogenesis, however, requires bi-allelic APC inactivation (i.e., mutations in both alleles) or silencing of one allele by loss of heterozygosity [7,8]. Most APC mutations found in sporadic or familial CRC patients are clustered in the middle part of the protein and are either nonsense or frameshift mutations that generate a premature stop codon and are responsible for the subsequent expression of a truncated inactive protein product [9,10]. Although APC has multiple cellular functions, the initiation of CRC is related to its role as a negative regulator of the canonical Wnt signaling pathway and

downregulation of the β-catenin oncogene [11,12]. APC mutations that promote aberrant activation of the Wnt pathway lead to increased cell proliferation and dysplastic adenomatous lesions [13]. Although CRC has been studied for many years, the treatment options remain limited [11], and additional, personalized, strategies are greatly needed.

Organoids are stem cell-derived 3-dimensional in vitro multicellular structures that exhibit structural, functional, and molecular similarity to the tissue of origin. Two main types of organoids can be defined based on the stem cells origin: organoids derived from pluripotent stem cells (PSCs) whether embryonic stem cells (ESCs) or induced PSCs (iPSCs), and organoids derived from adult stem cells (ASCs) [14]. Colonic organoids, or colonoids [15] can be grown from ASCs obtained from either normal colonic epithelium or neoplastic lesions [16]. Long-term expansion of epithelial colonic crypts was first demonstrated around a decade ago [17], when the leucine-rich repeat containing G protein-coupled receptor 5 (Lgr5[+]) was used to generate a continuously expanding, self-organizing epithelial structure from mouse intestinal stem cells that was reminiscent of the normal gut. LGR5 is a marker for intestinal stem cells and an essential factor for Wnt signaling activation in intestinal crypts [18]. Culture conditions enabling long-term expansion are based on the growth requirements of the intestinal epithelium. The Wnt agonist R-Spondin 1 was included in the intestinal culture, because Wnt signaling has a central role in crypt proliferation. Significantly, R-spondin 1 is also an LGR5 ligand. Epidermal growth factor (EGF) and Noggin, which are associated with intestinal proliferation and induced crypt expansion, are also included in the colon organoid growth supplements [19]. Further improvements and modifications of the conditions to promote long term culture are carried out in order to grow the colon organoids in laminin-rich Matrigel, which supports intestinal epithelial growth by replicating the enrichment of laminin in the crypt base [20], and the inclusion of the Wnt3a ligand in the culture medium to activate the canonical Wnt pathway, as well as inhibitors of Alk and p38 [17].

Here, we describe the establishment of a human colonoid system derived from intestinal crypts isolated from colonic biopsies obtained from polyposis patients (FAP and other genetic backgrounds, see Table 1). The isolated crypts form spheroid-like structures with the apical side towards the newly formed lumen, and the basolateral surfaces facing the Matrigel and medium.

**Table 1.** Colon organoids were established from patients with various genetic backgrounds.

| Patient/Isolation | Gender | Genetic Background | Growth (days) | | Structure (days) | |
|---|---|---|---|---|---|---|
| | | | **H** | **P** | **H** | **P** |
| 1-1 | F | APC ns S874X | 3 | 21 | U | B (1–15); C (15–21) |
| 1-2 | | | 43 | 43 | B (1–43) | B (1–15); C (15–43) |
| 2 | F | APC ns L77X | X * | 15 | X | U (1–15) |
| 3 | M | APC ns L77X | – | 20 | – | B&C (1–20) |
| 4-1 | M | APC ns Q341X | 7 | – | C (1–7) | – |
| 4-2 | | | 3 | 39 | X | B&C (1–39) |
| 5-1 | M | APC ns Q341X | 3 * | 14 | X | C (1–14) |
| 5-2 | | | 27 | – | C (1–27) | – |
| 6-1 | M | APC ns R302X | 6 * | 6 * | C (1–6) | X |
| 6-2 | | | 7 | 19 | B (1–7) | B (1–19) |
| 6-3 | | | 8 | 22 | B (1–8) | B&C (1–22) |
| 7 | F | APC fs Z1379R | 21 | 29 | C (1–21) | B&C (1–29) |
| 8 | F | APC fs Q264 | 5 | 19 | C (1–5) | B (1–19) |

**Table 1.** *Cont.*

| Patient/Isolation | Gender | Genetic Background | Growth (days) | | Structure (days) | |
|---|---|---|---|---|---|---|
| | | | **H** | **P** | **H** | **P** |
| 9-1 | | APC fs R332 | 4 * | 7 * | U | B (1–7) |
| 9-2 | | | 8 | 8 | B (1–8) | B (1–8) |
| 10 | F | APC fs 2688delC in exon 15 | 10 | – | B (1–10) | – |
| 11 | M | MYH G382D −/− | X | 39 | X | B (1–11); C (11–39) |
| 12 | | MYH 1145G>A +/− RAD50 326_329delCAGA +/− | 7 | 21 | B&C (1–7) | B&C (1–21) |
| 13 | M | POLD1 V759I +/− | – | 9 | – | 2D (7<) |
| 14 | | PTEN 697C>T +/− | – | 25 | – | 2D (23<) |
| 15 | M | Ulcerative Colitis | 7 | – | U (1–7) | – |
| 16 | M | Unknown | 3 | 36 | U | B (1–11); C (11-36) |
| 17 | | Unknown | X | 36 | X | B (1–18); C (18–36) |
| 18 | F | Unknown | 7 | 7 | B (1–7) | B (1–7) |

Ns = Nonsense, Fs = Frameshift, X = No growth, * = Contaminated, B= Budding, C = Cystic, U = Undefined, 2D = 2 dimensional, 1–10—FAP syndrome, 11–12—MAP syndrome, 13–18—polyposis/sporadic.

As already described, the APC protein plays a critical role in the maintenance of colonic epithelium homeostasis by negatively regulating the canonical Wnt pathway. Mutations in the APC gene are detected very early in the adenoma-carcinoma sequence and the APC protein is thought to act as a 'gatekeeper' for the development of colorectal carcinogenesis [7,21]. Approximately 30% of the mutations in FAP patients are single nucleotide changes that result in premature termination codons (PTCs). A promising therapeutic approach for the treatment of this subset of patients, is the use of nonsense mutation readthrough therapy that can restore the full-length length APC protein [22,23]. Here, we show that organoids established from adenomas of FAP patients with APC nonsense mutations can be used to test the feasibility of readthrough-based personalized intervention aimed to restore normal APC production and inhibit cancer initiation.

## 2. Materials and Methods

### 2.1. Human Tissue Samples

The study was approved by the ethics committee of Tel-Aviv Sourasky Medical Center and Tel Aviv University. All samples were obtained after patient informed consent was signed. Colonoscopy biopsy specimens of adenoma and healthy surrounding tissue were obtained from patients at the Sourasky Medical Center and kept in sterile PBS on ice before processing.

### 2.2. Isolation of Intestinal Crypts from Human Colorectal Biopsies

The isolation protocol is based on previous reports [17,24,25], with the following modifications: Freshly obtained samples were washed X5 in sterile PBS containing 50 µg/mL Gentamycin (Biological Industries, Beit haemek, Israel; 03-035-1C) and 100 µg/mL Primocin (Invivogen, CA, USA; ant-pm-1,termed: PBS-GP). The samples were then diced with a scalpel and incubated for 10 min in PBS-GP + 2.5 µg/mL Amphotericin B (AMB) on ice. Dispase (gentle protease, Corning; 40-235) was added to dissociate the crypts, and the samples were incubated on a rotary mixer for 1 h at 4 °C.

Manual isolation under the microscope: Dispase was removed carefully and 3 mL FBS was added to the samples for 15 min on ice. 3 mL of DMEM/F12 was then added to the samples followed by vigorous pipetting to isolate the crypts. The isolated crypts were manually isolated under a microscope with a P200 tip in sterile conditions and moved to

a clean Petri dish with DMEM/F12. All isolated crypts were moved to a falcon tube and centrifuged at $300 \times g$ for 3 min at 4 °C, supernatant was removed.

Shaking method: Dispase digestion was followed by three PBS-GP washes and vigorous shaking in PBS-GP. The isolated crypts were transferred to a Falcon tube containing 0.5 mL FBS, and centrifuged at $300 \times g$ for 3 min at 4 °C. The pelleted crypts were then resuspended in approximately 200 µL Matrigel (Corning, NY, USA; 354234) per sample and seeded as four 10 µL drops per well in a 24-well plate. The resultant Matrigel domes were overlaid with 500 µL LWRN complete medium (described below) supplemented with 10 µM Y-27632 (Stem cell; 72304), 100 nM SB431542 (Stem cell; 72232), 10 µM SB202190 (Sigma, Rehovot, Israel; S7067), 2.5 µM CHIR99021 (Stem Cell; 72052), and 2.5 µg/mL Amphotericin B (AMB). Medium was replaced every day for the first 3 days in culture, and then every other day. The media were replaced every other day. AMB was added to the medium only for the first 3 days, with CHIR added for the first 10 days of growth.

### 2.3. LWRN Complete Medium

LWRN complete medium is composed of 50% Human 2X Basal medium and 50% LWRN conditioned medium supplemented with EGF to 100 ng/mL (R&D Systems, MN, USA; 236-EG-200) and 100 µg/mL Primocin (InvivoGen; #ant-pm-1). Human 2X Basal medium is composed of Advanced DMEM/F-12 (Invitrogen, MA, USA; 12634028) supplemented with 4 mM Glutamax (Thermofisher, MA, USA; 25050-061), 20 mM HEPES (Biological Industries), 2% N2 supplement (Thermofisher; 17502-048), 4% B27 supplement (Invitrogen; 12587010), 2 mM N-acetylcysteine (Sigma; A9165-5G), and 0.5% PenStrep (Biological Industries, Beit-Haemek, Israel). The Human 2X Growth medium was prepared separately, filter sterilized through a 0.2 µm filter (Sartorius, Vienna, Austria; 16534-K), aliquoted, and frozen at −20 °C. LWRN conditioned medium was prepared from LWRN cells (ATCC, MD, USA; CRL-3276) according to the ATCC datasheet protocol. Conditioned media was filter sterilized using a 0.2 µm filter, and aliquots were frozen at −20 °C.

Additional Supplements: Noggin (Peproteck, Rehovot, Israel; 250-38-20) and R-Spondin (Peproteck 120-38-20) were used as a supplement to the growth medium shown in Figure S1.

### 2.4. Cryopreservation and Thawing of Colon Organoids

Protocols were based on [26] with slight modifications. Freezing: Briefly, medium was discarded from the Matrigel domes containing colon organoids. Matrigel domes were scraped from the wells with a cut tip and washed in 2 mL cold PBS. 2 mL PBS containing Matrigel domes was moved to a conical tube. Organoids were disengaged from the Matrigel by $10 \times$ triturating. PBS was added up to 10 mL and organoids were pelleted by centrifuging at $300 \times g$ for 3 min at 4 °C. Supernatant was discarded and pellet resuspended in 80% LWRN complete media and 10% FBS. The resuspended organoids were transferred to a pre-cooled freezer vial, and 10% DMSO was added and swirled in the tube. The tube was immediately moved to −80 °C for 24 h and then to −150 °C for long-term cryopreservation. Thawing: Hand-thaw till first evidence of thawing, then keep vial on ice. Transfer contents of vial to conical tube with ice-cold 80% DMEM/F12 + 20% FBS. Pellet the organoids by centrifuging at $300 \times g$ for 3 min at 4 °C. Discard the supernatant and resuspend the pellet carefully with Matrigel 8 mgr/mL, breaking up the colonoids and creating a homogenous mixture with a precooled tip. Seed 10 µL drops in a 24-well plate, 4 drops per well. Overlay with 500 µL LWRN compete medium supplemented with 10 µM Y-27632 (Stem cell; 72304), 100 nM SB431542 (Stem cell; 72232), 10 µM SB202190 (Sigma; S7067), 2.5 µM CHIR99021 (Stem Cell; 72052). CHIR was added only on the day of thawing.

### 2.5. Immunofluorescence (IF)

IF assays were conducted as previously described [25], with all steps at room temperature (RT) unless otherwise stated. All materials were diluted in PBSX1 unless noted. Briefly: Colon organoids in Matrigel were transferred to 15 µ-Slide 8 well plates (Ibidi, Martinsried,

Germany; 80826), and allowed to grow for several days. After this time, the medium was discarded and the Matrigel domes containing the organoids were washed X3 in PBSX1. Samples were incubated in 4% PFA (EMS, PA, USA; 15710)/PBS for 20 min followed by incubation in 20 mM glycine (Sigma; G7126) for 10 min. The samples were then washed X3 in PBSX1, permeabilized in 0.2% Triton X-100 (Sigma) for 30 min, and incubated in blocking buffer (1% BSA (Amresco, OH, USA; 0332-TAM), 0.1% Triton X-100) for 45 min. Primary antibodies were added to the samples in blocking buffer at a final volume of 100 μL and incubated at 4 °C overnight. The following primary antibodies were used; rat anti E-Cadherin 1:200 (Millipore, Walford, UK; MABT26), mouse anti Mucin2 1:200 (Santa Cruz, CA, USA; sc-515032), rabbit anti Ki67 1:250 (Abcam, Cambridge, UK; ab15580), rabbit anti Cleaved Caspase-3 (Cell Signaling, MA, USA; (Asp175) #9661). After incubation, the samples were washed X3 with 0.2% Triton X-100 and incubated with secondary antibodies (all from Invitrogen) in blocking buffer at a final volume of 100 μL. The secondary antibodies included Alexa fluor 488 goat anti-mouse 1:500 (A1101); Alexa fluor 488 goat anti-rabbit 1:500 (A11034), Alexa fluor 488 rabbit anti-rat 1:500 (A21210), and Alexa fluor 647 goat anti-mouse 1:500 (21236). The incubated samples were washed X2 with 0.2% Triton X-100 and the cell nuclei were stained with 10 μg/mL 4′,6-Diamidino-2-phenylindole (DAPI; Sigma) for 5 min in the dark. Samples were washed again, mounted (GBI labs, WA, USA; E18-18), and dried overnight.

### 2.6. Microscopy and Imaging

Bright field pictures of the colonic organoids were taken using the FLoid Cell Imaging Microscope (Life technologies, Herzliya, Israel). Fluorescent pictures were taken by a Zeiss LSM700 or LSM800 confocal laser scanning microscope (Carl Zeiss MicroImaging, Jena, Germany).

### 2.7. RNA Isolation and RT-qPCR Analysis

The following protocol was calibrated for 10 μL droplets seeded in 96 wells. Matrigel domes containing colonoids were scraped from the wells using a cut tip, washed with 200 μL PBS-PG and transferred to an Eppendorf tube containing 200 μL PBS-GP on ice. Colonoids were harvested from the Matrigel by pipetting X30. PBS-PG was added to a final volume of 1 mL, and the colonoids were pelleted by centrifuging at $300 \times g$ for 3 min at 4 °C. The supernatant was discarded and 200 μL Trizol Reagent (Bio-lab, Jerusalem, Israel) was added to the colonoid pellet, followed by vigorous pipetting to extract the RNA. The samples were then centrifuged at $12,000 \times g$, 4 °C for 15 min, the supernatant was transferred to fresh tubes, 40 μL chloroform was added, and the samples were incubated for 15 min at R.T. Centrifuging at $10,000 \times g$ for 15 min at 4 °C, separated 3 phases. The upper (clear fluid) layer, which contains the RNA was transferred to a new Eppendorf tube and 500 μL isopropanol was added followed by 5–10 min incubation at R.T and the addition of 1 μL glycogen. The samples were incubated overnight at −20 °C and then centrifuged at $12,000 \times g$ for 12 min at 4 °C. The RNA pellet was washed X3 with 1 mL 75% pre-cooled EtOH, followed by centrifugation at $7500 \times g$ for 5 min at 4 °C. It was then air-dried and solubilized in nuclease-free water. The isolated RNA was used to generate complementary DNA (cDNA) using the iScript cDNA synthesis kit (Bio-Rad, Rehovot, Israel) according to the manufacturer's instructions. RT-qPCR analysis was performed using SYBR Green Master mix (PCR Biosystems London, UK) following the manufacturer's instructions. qPCR was carried out using the C1000 Touch thermal cycler. All reactions were in triplicates. GAPDH or Actin were used to normalize target gene expression.

### 2.8. qPCR Primers

Primers were planned using the PrimerBank and were ordered from Sigma (Rhovot, Israel) with synthesis scale of 0.025 μmole, dry and desalted (Table 2).

**Table 2.** Primer sequences.

| Gene Symbol | Fw Primer | Rv Primer |
| --- | --- | --- |
| GAPDH | GCACCGTCAAGGCTGAGAAC | ATGGTGGTGAAGACGCCAGT |
| MYC | CTCGGTGGTCTTCCCCTACCCT | TGTCCAACTTGACCCTCTTGGC |
| AXIN2 | GCAGCTCAGCAAAAAGGGAAAT | TACATCGGGAGCACCGTCTCAT |
| CCND1 | CATCTACACCGACAACTCCATC | TCTGGCATTTTGGAGAGGAAG |
| SOX9 | ACTTGCACAACGCCGAG | CTGGTACTTGTAATCCGGGTG |
| LEF1 | AGACAAGCACAAACCTCTCAG | TCATTATGTACCCGGAATAACTC |
| ACTIN | CCTGGCACCCAGCACAAT | GGGCCGGACTCGTCATACT |

*2.9. Western Blot Analysis (WB)*

The following was calibrated for 10 μL droplets seeded in 96 wells. Matrigel domes containing colonoids were scraped from the wells using a cut tip, washed with 200 μL PBS-PG and transferred to an Eppendorf tube on ice. Colonoids were disintegrated from Matrigel by pipetting ×30 and PBS-PG was added to a final volume of 1 ml. Colonoids were pelleted by centrifuging at $300\times g$ for 3 min at 4 °C. The supernatant was discarded, and the colonic organoid pellet was lysed in 75 μL ice-cold Lysis buffer (20 mm Tris-HCl pH 7.4, 150 mm NaCl, 1% NP-40, 1% Triton X-100, 0.05% SDS, 10% glycerol) containing protease inhibitor cocktail ×1 (Sigma). Lysates were run on SDS–PAGE gels and transferred to nitrocellulose membranes. The membranes were blocked with 5% skim milk in PBST (0.1% Tween) and incubated overnight at 4 °C with primary antibodies, then for a further 60 min with secondary antibodies, and were then subjected to chemiluminescence analysis. The following antibodies were used for Western blot (WB) analysis. Primary antibodies: rabbit anti non-phospho (active) β-catenin 1:1500 (Cell signaling; 19807S); rabbit anti APC 1:500 (Santa Cruz; sc-7930); mouse anti-Actin 1:10,000 (ImmunO; 69100). Secondary antibodies: goat anti-mouse IgG and goat anti-rabbit IgG 1:10,000 (Jackson ImmunoResearch, PA USA).

*2.10. Statistical Methods*

Data were analyzed with GraphPad Prism software (version 9.0, GraphPad, La Jolla, CA, USA) and the results are presented as the mean with standard deviation of 3–5 repeats. An unpaired *t*-test or analysis of variance (ANOVA) was used to assess the significance of variations; multiple comparisons were conducted according to software recommendations.

**3. Results**

*3.1. Establishing and Characterizing Adenoma-Derived Colonoids*

Adenoma biopsies were obtained from polyposis patients undergoing routine surveillance colonoscopies in the gastroenterology department. Colonic crypts were isolated from the samples by dissociation with Dispase (protease used for gentle dissociation) and vigorous shaking in PBS. Crypts were seeded in laminin-rich Matrigel and grown in a medium containing Wnt3a, R-Spondin, Noggin, EGF, and additional factors that support colon organoid (colonoid) long-term growth and expansion [24,25]. Bright-field images demonstrate colonoid growth and development (Figure 1A). Prolonged growth (over 20 days) resulted in spherical colonoids, a form reflecting decreased differentiation and diminished budding morphology as previously described by Sato et al. [17], and is typically seen in colonoids grown in Wnt containing medium [27] (Figure S1). We used a variety of specific cellular markers to characterize the differentiated cells making up the colonoid structure. Specifically, E-Cadherin was used to identify mature epithelial colon cells (Figure 1B) and the Mucin2 marker was used to label goblet cells (Figure 1C). Proliferating cells, identified by the Ki67 marker, were present throughout the colonoid structure after prolonged growth (20–50 days), confirming that the organoids are regenerating structures (Figure 1D). Finally, clearance of cells from the lumen was detected by the apoptosis marker, Cleaved

Caspase-3 [28] (Figure 1E). This resembles the physiological process in which dead cells are exfoliated from the colon lumen. These results demonstrate that colonic organoids derived from polyposis patients with various genetic backgrounds (Table 1) contain mature colon cell types and maintain organ topology.

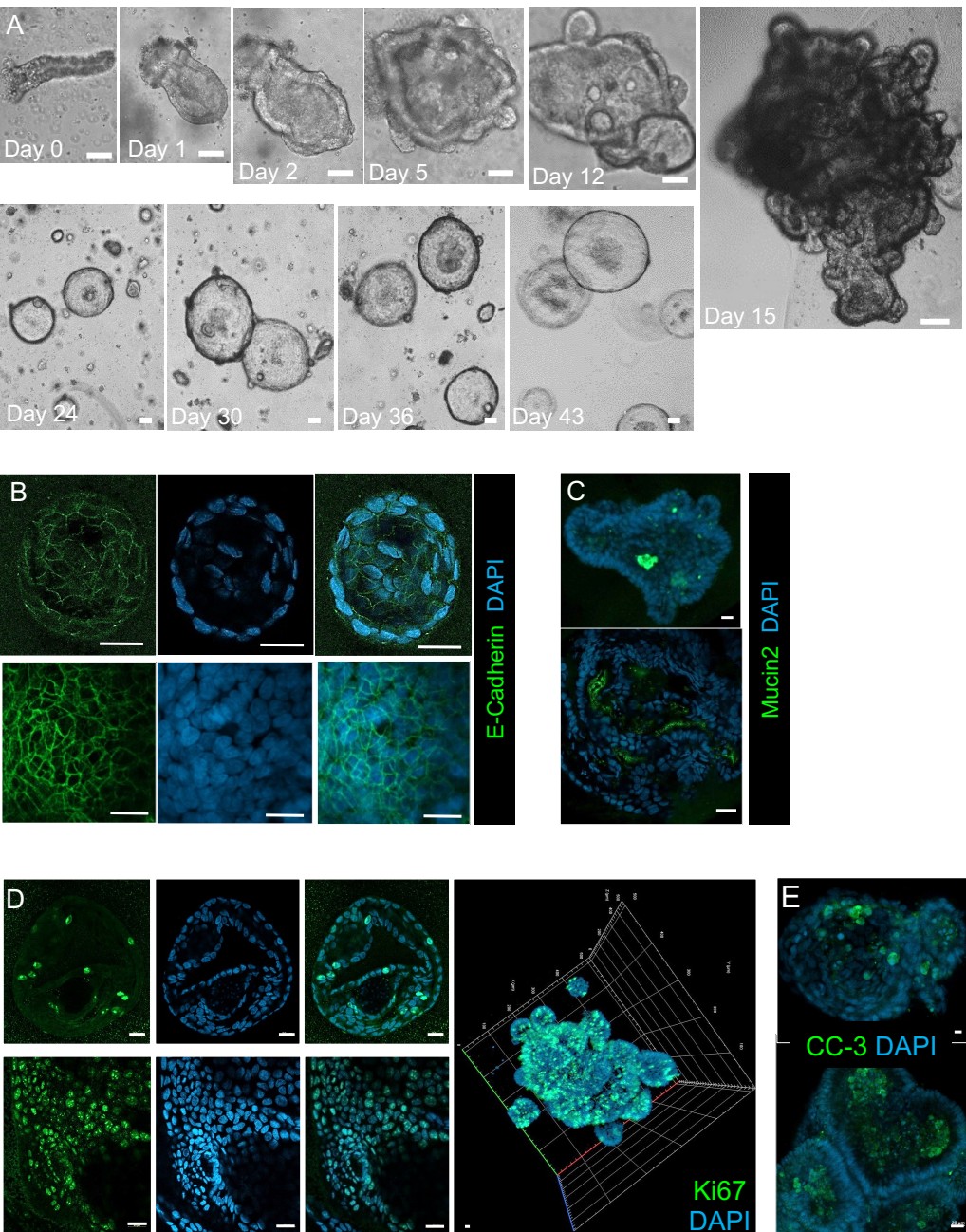

**Figure 1. Establishing and characterizing adenoma-derived colonoids**. (**A**) *Representative colonoid cultures*: Polyp-derived colonoids were established from patient biopsies. Crypts were extracted as described, embedded in Matrigel and submerged in supplemented media. Colonoids cultured and expanded for 10–15 days formed budding structures with an internal lumen, but subsequently became steadily more cystic surviving for an additional 25–30 days. Bright-field images were taken at 20 Magnification. Scale bar, 125 μm (**B**–**E**) *Immunofluorescence analysis of polyp-derived colonoids*: Colonoids were grown as in (**A**) and subjected to fluorescent staining to detect E-Cadherin (epithelial cell marker; (**B**)), Mucin2 (goblet cells; (**C**)), and Ki67 (proliferating cells; (**D**)). Apoptotic cells in the colonoid lumens were identified by the Cleaved Caspase 3 marker (**E**). Scale bar, 20 μm.

### 3.2. Enhancing the Growth of Healthy Tissue-Derived Organoids

Most colonoids grown from adenoma tissues expended more efficiently compared to colonoids derived from surrounding healthy samples (Figure 2A). In several experiments (using samples from the same patient) adenoma-derived colonoids survived longer exhibiting higher evidence of renewal, with more budding structures, compared to colonoids derived from healthy tissue which became spherical or degraded in culture and stopped growing or developing after approximately 14 days (Figure 2B). Increasing the concentration of LWRN conditioned medium containing Wnt3a, Noggin and R-Spondin as well as the EGF levels in the growth medium of the healthy tissue-derived colonic organoids, prolonged the duration of their growth and development (Figure 2C).

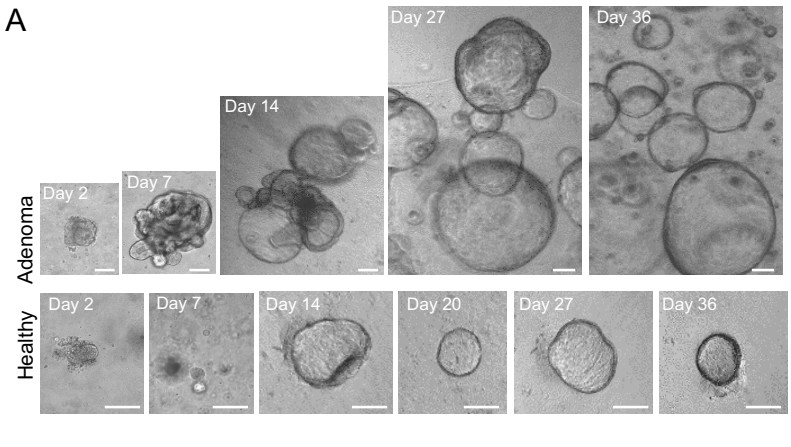

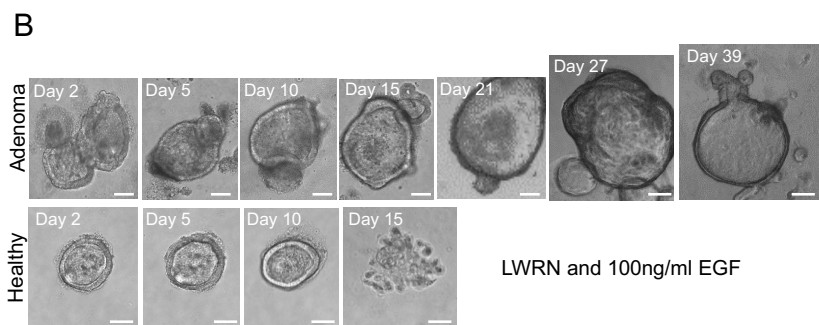

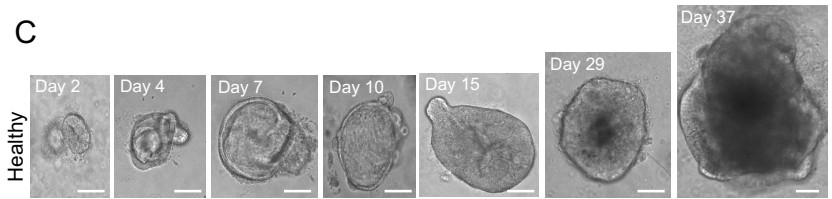

**Figure 2. Enhancing the growth of healthy tissue-derived organoids**. (**A**) Representative colonoid cultures derived from healthy tissue or polyp biopsies. The colonoids were grown in LWRN complete medium (see Section 2 Material and Methods), which supports only limited expansion of the healthy tissue-derived colonoids, Scale bar, 125 μm (**B,C**) Colonoids derived from a polyp or surrounding healthy tissue samples were grown in LWRN complete medium (**B**) or enriched medium containing a higher concentration of LWRN (X2 total) and 200 ng/mL EGF (**C**). Bright-field pictures were taken to monitor colonoid development, Scale bar, 125 μm.

Modifications were also made to the crypt isolation protocol in order to optimize our colonic organoid culture of both adenoma and healthy tissue samples. Specifically, instead of manually isolating the crypts under a microscope, the tissue was dissociated and vigorously shaken to release the crypts. This greatly increased the number of crypts (Figure S2) that could be cryopreserved and re-established in culture (Figure S3).

*3.3. G418 Induces APC Restoration Decreasing Canonical Wnt Signaling and Cell Proliferation in Treated Colonoids*

In order to test the feasibility of nonsense mutation readthrough and establish proof-of-principle for our model system, we tested the effect of G418, a potent antibiotic known to enhance nonsense mutation readthrough, on adenoma-derived colonoids, obtained from patients harboring a germline APC nonsense mutation. WB analysis revealed that APC expression was partially restored following treatment (Figure 3A,D). We have previously shown that the administration of Erythromycin (another documented readthrough compound), in order to restore APC expression in FAP patients harboring APC nonsense mutations, reduced the adenoma burden in most patients tested [22]. However, patients' responses to the treatment varied, and appeared to be affected by additional factors besides the specific APC nonsense mutation. As depicted in, Figure 3B; left panels, the effect of nonsense mutation readthrough treatment on Wnt target genes was extremely variable among the treated patients, as evidenced by the RT-qPCR analysis of adenoma and healthy tissue samples obtained from these patients. Thus, there remains a need for new tools that may guide precise intervention strategies and predict the therapeutic outcome. APC plays a key role in the regulation of the canonical Wnt signaling pathway by reducing β-catenin levels [7]. Indeed, as shown in Figure 3C (quantitation in Figure 3D), G418 treatment decreased the levels of non-phosphorylated (active) β-catenin, and reduced the expression of several Wnt target genes (Figure 3B; right panels). The treatment had no significant effect on either APC or active β-catenin expression in healthy tissue derived colonoids (Figure S4). Importantly, G418 treatment also reduced proliferation in the colonoid sample (polyp-derived) as detected by immunofluorescence assay of the Ki67 marker (Figure 3E; upper panel). The decrease was not observed in the healthy tissue-derived colonoid where APC is still partially expressed (Figure 3E; lower panel). A similar decrease in proliferation was observed in the majority of the patients who participated in our clinical trial who were treated with Erythromycin [22]. Representative immunohistochemistry assays performed on biopsy samples obtained from the same patient revealed that robust Ki67 staining in the adenoma samples, was significantly decreased following readthrough treatment (Figure 3F). These results suggest that additional factors also affect Wnt signaling levels in vivo. Therefore, the colonoids may be used to test different treatment strategies aimed at reducing cell proliferation which can be easily monitored by examining the levels of both CyclinD1 and Ki67.

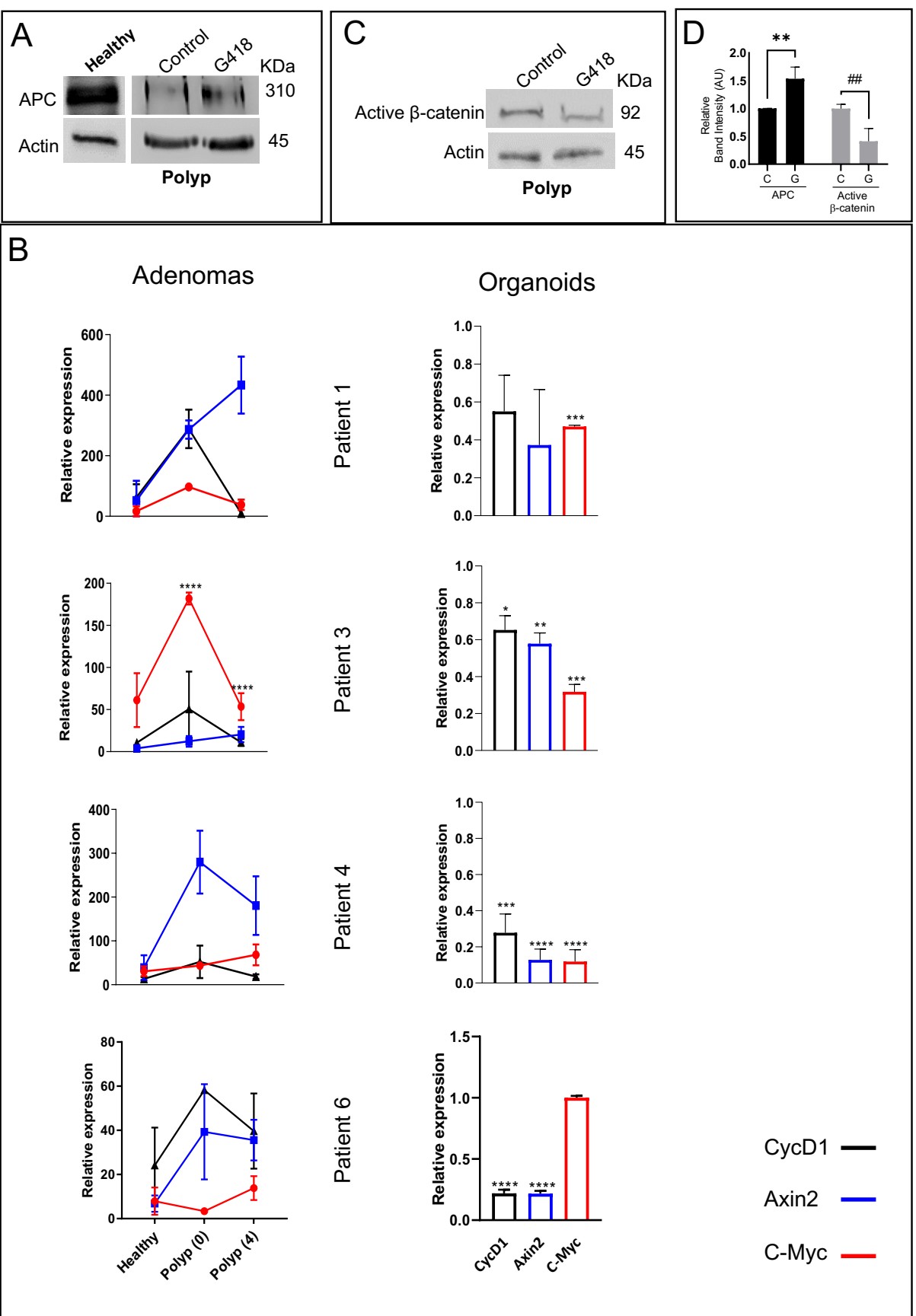

**Figure 3.** *Cont.*

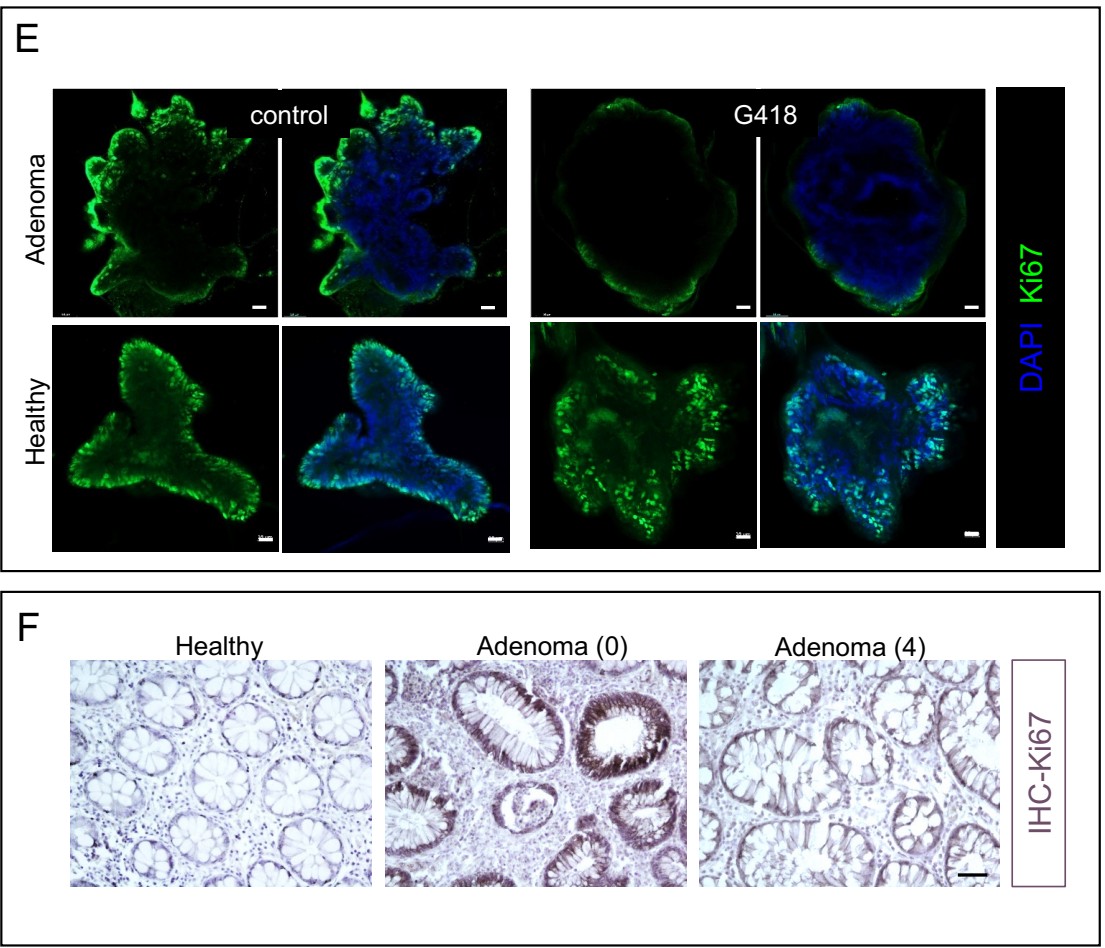

**Figure 3. Induced APC expression in colonoids decreases cell proliferation and canonical Wnt signaling**. Polyp-derived colonoids from patient 1 (S874X), patient 3 (L77X), patient 4 (Q341X) and patient 6 (R302X) were treated with G418 (150 µg/mL) for 3–6 days. (**A**) Total protein was harvested and subjected to WB analysis using the indicated antibodies (a revpresentative Western blot of colonoids derived from patient 4; healthy and adenoma tissue). (**B**) **Left**: FAP patients 1, 3, 4 and 6 were previously subjected to 4 months of erythromycin treatment. Samples were obtained pre-trial (0) and post-trial (4). Polyps were removed from each patient along with a sample of healthy tissue. RNA was extracted using the AllPrep DNA/RNA/protein kit (QIAGEN) and RT-qPCR was performed using validated primers. Two-way ANOVA with Tukey's multiple comparisons test was applied, **** $p < 0.0001$. **Right**: Total RNA was extracted from polyp-derived colonoid samples (patients 1, 3, 4 and 6), converted to cDNA, and subjected to RT-qPCR analysis. Transcript levels of canonical Wnt target genes were analyzed, comparing their levels prior to and following G418 treatment. Two-tailed *t*-test was applied ($\alpha = 0.05$). Patient 1: *** $p = 0.0001$. Patient 3: * $p = 0.0014$, ** $p = 0.0002$, *** $p < 0.0001$. Patient 4: *** $p = 0.0003$, **** $p < 0.0001$. Patient 6: **** $p < 0.0001$. (**C**) Total protein was harvested from treated colonoids and subjected to WB analysis using the indicated antibodies (a representative Western blot of colonoids derived from patient 1). (**D**) Quantification of protein expression levels relative to actin, following G418 treatment using band intensity (N = 3). Two-way ANOVA was employed (*p*-value = 0.0003, with Šídák's multiple comparisons test: ** $p = 0.00$ and ## $p = 0.007$. (**E**) Colonoids were established from patient 6 (R302X; adenoma; upper and healthy tissue; **Lower Panels**) and treated with 25 µg/mL G418 for 2 days. Proliferating cells were identified by the Ki67 marker, and cell nuclei were counterstained with DAPI. *p* = P-value; Scale Bars, 20 µm. (**F**) PFA-fixed sections of adenoma or healthy tissue samples were obtained from FAP patient 6 as part of our clinical trial. Samples were taken pre-trial (0) and post-trial (4), stained for the proliferation marker Ki67, and images were taken by light microscopy (×20).

## 4. Discussion

Organoids, which represent an accurate 3-dimensional colonic model, can improve our ability to study the development, physiology, and pathology of the colon as well as assist in the development of new therapeutic modalities [29–31].

The organoid model system was first introduced over a decade ago, and besides advancing studies in biology and biomedicine, has made an increasing clinical impact on personalized medicine. Organoids are self-organized, multicellular structures generated from stem cells [32,33], either embryonic or induced pluripotent stem cells (together referred to as pluripotent stem cells, or PSCs) or adult stem cells (ASCs) [34,35]. Although the number of published studies in the field has been growing rapidly, information about organoids derived from ASCs is relatively scarce, and very little is known about colonic crypts isolated from pre-cancerous adenomas in polyposis patients. Studies concerning in vitro cultures of human primary intestinal tissue described various protocols and different supplements aimed to optimize colonoid growth and expansion [17,25,36]. Nevertheless, the precise conditions for growing such organoids in terms of crypt isolation and culture media are still being tested [37]. Here, we describe the establishment of an ASC human colonic organoid system derived from adenomas and surrounding healthy tissue obtained from various polyposis patients. The advantages of using adult tissue to establish a colonoid system, include the ability to use both healthy and adenoma tissue to compare developmental cues, as well as the opportunity to study individually personalized therapeutic strategies. However, there are a number of limitations to this technique, including the minute sample size, which reduces the number of colonoids formed, somatic as well as hereditary differences between donors, and high contamination rates. In the current study, we addressed several of these issues and demonstrated improvements in the colonoid cultures derived from both adenoma samples and the surrounding healthy tissues. Our results revealed that digesting the tissue with Dispase reagent and loosening the crypts by shaking, enabled us to isolate a large number of crypts from small amounts of tissue biopsy (Figure S2). This method improves the efficiency of crypt isolation and promotes the expansion and prolonged budding of colonoids. Colonoids were derived from polyposis patients with different genetic backgrounds including germline APC mutations (Table 1). In general, colon organoids derived from adenoma tissue thrived for longer periods of time in culture and developed more budding and cystic structures than those derived from healthy tissue (Figure 2, Table 1). However, we were able to prolong growth and improve the development of colonoids derived from healthy tissue to a comparable level to adenoma-derived organoids by supplementing the culture medium with EGF and LWRN conditioned medium (Figure 2B, lower panel). We were also able to improve cryogenic preservation and recover colonoids from frozen samples (Figure S3), enabling us to create a reservoir of samples obtained from patients with rare genetic backgrounds. One of the advantages of colonoids derived from ASCs is that they exploit the relevant physiological tissue regeneration processes. Thus colon organoids grown from adult tissue samples are entirely epithelial structures and contain the mature epithelial cell types and topological structure of the colon [38]. Accordingly, we detected goblet cells and epithelial cells [39] in our colonoid system. Together with the proliferating cells (Ki67) detected throughout the colonoid structure and the dead cells (Cleaved caspase 3) identified in the colonoid lumen [17], the evidence suggests that our cultured organoids can accurately mimic the cellular and architecture of their colonic origin including their pre-malignant traits. As such, the adenoma-derived colonoids may serve as a convenient model system for testing a patient-specific therapeutic intervention.

APC germline mutations lead to the development of FAP, manifested by multiple colonic adenomas, which will progress into CRC if not removed. In fact, mutations in the APC tumor suppressor gene are found in over 80% of all hereditary and sporadic CRC syndromes [40]. APC downregulates the expression levels of the oncogene β-catenin, and thus functions as a negative regulator of the canonical Wnt pathway [7]. Around 30% of the APC mutations in FAP patients are nonsense mutations, which generate premature

termination codons and result in the expression of a nonfunctional protein [41]. Certain antibiotics such as aminoglycosides can alter ribosomal activity in a way that enables readthrough and thus the production of the complete protein [42]. However, although nonsense mutation readthrough initially appeared to have potential as a novel therapeutic strategy for genetic diseases with nonsense mutations in key genes, aminoglycosides are toxic and cannot be consumed for long periods. A more precise personalized approach, designed to identify potent readthrough agents for specific mutation types, may greatly improve this field of therapy. Here, we present proof-of-concept of the nonsense suppression capabilities of the known readthrough inducing aminoglycoside-G418 applied to our colonoid system. Treatment with G418 partly restored APC protein expression, reducing both proliferation and active β-catenin protein levels, and decreasing the transcription of several Wnt target genes, thereby inhibiting the oncogenic canonical Wnt pathway. These results suggest that colonoids grown from human adult adenoma tissue may be used for testing various readthrough agents and that this system, in addition to the FAP-iPSCs–derived organoids [43], may be used to develop personalized drug regimens and therapeutic approaches similar to other organoid types [33].

**Supplementary Materials:** The following are available online at https://www.mdpi.com/article/10.3390/organoids1010006/s1, Figure S1: Organoids derived from Adenomas grown in different media, Figure S2: Adjusting crypt isolation method. Figure S3: Cryogenic Preservation. Figure S4: G418 treatment of Healthy tissue-derived colonoids.

**Author Contributions:** T.E.-L.—performed the vast majority of the experiments and analysis and edited the manuscript, M.C.—helped in experimental design and manuscript preparation, A.W. and Y.S.-A.—performed part of the experiments, O.S. and K.H.—helped with the imaging, R.K.—provided the clinical samples and helped in experimental design and manuscript preparation, R.R.-A.—oversaw the project, analyzed the data, and prepared the manuscript. All authors have read and agreed to the published version of the manuscript.

**Funding:** This research was funded by SPARK TEL-AVIV and the Foundation Jérôme Lejeune (Grant number 1727).

**Institutional Review Board Statement:** The study was conducted according to the guidelines of the Declaration of Helsinki, and approved by the Institutional Review Board (or Ethics Committee) of Tel-Aviv University (protocol code: 0667-20-TLV, date of approval: 13 October 2020).

**Informed Consent Statement:** Informed consent was obtained from all subjects involved in the study.

**Data Availability Statement:** Not applicable.

**Conflicts of Interest:** The authors declare no conflict of interest.

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
