# Peer review of "Adenoma-Derived Organoids for Precision Therapy"

_2674-1172, doi:10.3390/organoids1010006_

Round 1

Reviewer 1 Report

Dear Authors,
congratulations for this very interesting paper. The work traces a path towards standardizing innovative personalized therapies for patients affected by familial colorectal polyposis. The experimental design is simple and correct and the experiments are correctly carried out. The conclusions are supported by the results obtained and the work is written in a correct form and easy to read.
I suggest the authors review only the figures' legends, which in some cases do not clarify all the symbols shown in the images. Specifically, in fig 3, what does the abbreviation "Conc" mean? In the same figure the authors should review the right part of box B. What are the samples?

Best regards

Reviewer 2 Report

Dear authors, 

The present study deals with an interesting and relevant question, both methodologically and in terms of content. This should be seen in particular as, with an almost 100% risk of carcinoma, only prophylactic proctocolectomy is currently available as a "therapeutic" option in the presence of FAP.

Nevertheless, there are a few shortcomings in this work that need to be improved: 

  • The work of Crespo et al., 2017 (colonic organoids derived from human induced pluripotent stem cells for modeling colorectal cancer and drug testing, Nat Med, 23(7):878-884) is missing as one of the references. This should be included as a reference, as it also describes, albeit methodologically differently, the establishment of FAP organoids starting from iPCs and also a utilisation of these organoids for the investigation of a therapeutic window for, among others, aminoglcoside antibiotics inducing read through. 
  • Figure 3 A
    • In the continuous text, the treatment of 3 patients/organoids is described. However, only one sample is shown in the Western blot. If possible, a statement should be made as to whether the other samples provide a comparable result or the data could be shown in the supplemental. (The same applies to Figure 3 C)
    • It would be nice to show as well the "healthy" organoids treated with G418 to exclude a comparable effect in these wild type ones.
    • It would be nice to have the western blot results quantified
  • Figure 3 B: For easier understanding, it should be mentioned/shown here in comparison to which baseline a significant or non-significant reduction in Wnt target gene expression takes place - over time to the initially untreated situation? in comparison to treated "healthy" organoids?
